# COVID-19 in the Eastern Brazilian Amazon: Incidence, Clinical Management, and Mortality by Social Determinants of Health, Symptomatology, and Comorbidities in the Xingu Health Region

**DOI:** 10.3390/ijerph20054639

**Published:** 2023-03-06

**Authors:** Eric Renato Lima Figueiredo, Márcio Vinicius de Gouveia Affonso, Rodrigo Januario Jacomel, Fabiana de Campos Gomes, Nelson Veiga Gonçalves, Claudia do Socorro Carvalho Miranda, Márcia Cristina Freitas da Silva, Ademir Ferreira da Silva-Júnior, João Simão de Melo-Neto

**Affiliations:** 1Institute of Health Sciences, Federal University of Pará (UFPA), Belem 66075-110, Brazil; 2Faculty of Medicine of São José do Rio Preto (FAMERP), São José do Rio Preto 15090-000, Brazil; 3Laboratory of Epidemiology and Geoprocessing of Amazon, University of the state of Pará (UEPA), Belem 66050-540, Brazil; 4Faculty of Medicine, Federal University of Pará, Altamira 68370-000, Brazil

**Keywords:** COVID-19, incidence, social determinants of health, health services, health policy, pandemic

## Abstract

This study aims to investigate the relationship between social determinants of health (SDH), incidence, and mortality to verify which sociodemographic factors, symptoms, and comorbidities predict clinical management; second, this study aims to conduct a survival analysis of individuals with COVID-19 in the Xingu Health Region. Consequently, this study adopted an ecological framework, employing secondary data of COVID-19-positive individuals from the Xingu Health Region, Pará State, Brazil. The data were obtained through the database of the State of Pará Public Health Secretary (SESPA) for the period from March 2020 to March 2021. The incidence and mortality were higher in Vitória do Xingu and Altamira. Municipalities with a higher percentage of citizens with health insurance and higher public health expenditure showed a higher incidence and mortality. A higher gross domestic product was associated with a higher incidence. Females were found to be associated with better clinical management. To live in Altamira was a risk factor for intensive care unit admission. The symptoms and comorbidities that predicted worse clinical management were dyspnea, fever, emesis, chills, diabetes, cardiac and renal diseases, obesity, and neurological diseases. There were higher incidence, mortality, and lower survival rates among the elderly. Thus, it can be concluded that SDH indicators, symptomatology, and comorbidities have implications for the incidence, mortality, and clinical management of COVID-19 in the Xingu Health Region of eastern Amazonia, Brazil.

## 1. Introduction

Coronavirus disease 2019 (COVID-19) is a highly contagious infectious disease caused by a new betacoronavirus belonging to the large viral family of Coronaviridae. The first cases of COVID-19 were reported in December 2019 in Wuhan, China [1,2]. Initially, the outbreak of severe acute respiratory syndrome by type 2 coronavirus (SARS-CoV-2) was confirmed in the province of Hubei; however, this virus spread rapidly to several countries, causing a pandemic in 2020 [3,4,5].

Regarding symptoms, the clinical manifestation of COVID-19 virus can range from asymptomatic to severe [6]. The main clinical symptoms of COVID-19 patients include fever, cough, myalgia or fatigue, and dyspnea [7,8]. Additionally, olfactory and gustatory dysfunctions are common clinical findings in these patients [9]. Minor symptoms include sputum production, headache, haemoptysis [8], dizziness, diarrhea, nausea, vomiting [6], and skin lesions [10,11]. The fatality rate is approximately 5% (95% CI (0.01–0.11)) [12]. The prevalence of comorbidities is considered a risk factor for severe patients. However, the symptoms of infected patients are nonspecific [6], and there is a need to know the characteristics of each population and its implications for clinical evolution.

Since the start of the pandemic till the present, the literature on possible treatments for COVID-19 disease is increasing; however, vitamin supplements, anti-inflammatory agents, and antimicrobial therapy have shown a lack of efficacy in the treatment of patients; the best care strategy throughout the course of the disease remains unknown [13,14].

SARS-CoV-2 has negative effects on clinical practice. Regarding health workers, they are at higher risk of infection due to their efforts to protect the community; consequently, they are exposed to psychological distress, fatigue, and stigma [15]. Furthermore, in the initial phase of the COVID-19 pandemic, the mental health and well-being of the general population were affected, with increasing rates of suicidal thoughts among the population [16]. In addition to host factors, environmental or social factors contribute to the high infection risk, especially factors such as poor living conditions, nutrition, ventilation, sanitation, and overcrowding [17]. Although specific population groups may have higher risk factors, the differences in social-epidemiological patterns, and differences by age and gender have been little studied. Social inequalities due to different living and working conditions and socioeconomic status should be taken into account when assessing the risk to different population groups [18].

The Brazilian Amazon represents approximately 58.9% of Brazil’s national territory, hosts a complex biodiversity which includes diverse cultures, ecosystem services, and human settlements, with various degrees of urbanization and rurality ranging from metropolitan regions such as Belém and Manaus to traditional riverside, indigenous, and quilombola communities [19]. In the eastern Brazilian Amazonia, nine municipalities along the Transamazonian highway comprise the microregion of the middle Xingu. The Xingu River, one of the main right-bank tributaries of the Amazon Basin and the largest fluvial system in the world, runs alongside these municipalities [20]. The population of the state of Pará exceeds 370,000. Altamira is the most populous city in the region, with a population of 116,000 individuals. The other eight municipalities of the state include Anapú, Medicilândia, Vitória do Xingu, Brasil Novo, Senador José Porfírio, Porto de Moz, Pacajá, and Uruará (with populations between 10,000 and 50,000 inhabitants) [21].

The Xingu Health Region is one of the more than 400 health regions in Brazil established by states in collaboration with municipalities under the provisions of Decree No. 7508 of 2011 [22]. A health region is a geographic area composed of neighboring municipalities delimited by common characteristics, such as cultural, economic, and social aspects; additionally, in a health region, there is an integration of infrastructure and transportation to conduct health-related actions and deliver services in an integrated and equitable manner [22,23].

The Belo Monte Hydroelectric Power Plant, the largest of its kind in Brazil, is located in this region. This hydroelectric dam was constructed between 2010 and 2017 through the Growth Acceleration Program funding, which led to a transformation in the social and demographic profile of all municipalities in the area, including a population increase, as well as financial and commercial movements, resulting in numerous investments [20,24]. However, simultaneously, there has also been an increase in socioeconomic problems such as violence, exacerbation of agrarian conflicts, relocation of traditional populations, prostitution, and deficiency in educational and health services [25]. Thus, the multiplicity and territorial and cultural complexities of the Xingu region result in a scenario of great social vulnerability, which potentially has a significant influence on the transmission, morbidity, and mortality rates of COVID-19 [26,27].

As humanity’s coexistence with the pandemic evolves, there is a need to understand its legacy across different social organizations and populations affected by the health, environmental, and economic crises it has brought about globally, taking into account the local and regional aspects [28,29].

In this sense, the Brazilian health policy works to improve the living conditions and environment of the population by adopting a technical, operational, and organizational approach to the management of health interventions and services. This approach is based on the conceptual model of the social determinants of health (SDH) theory [30], which suggests that health and disease processes are influenced by factors arranged at different levels, from micro factors such as hereditary factors, age, gender, and lifestyle, to macro factors related to environmental, cultural, and socioeconomic issues [27,31,32].

Additionally, in a pre-vaccination context, understanding the clinical symptoms and comorbidities in each population and their implications for COVID-19 incidence and mortality can contribute to the construction of a more effective and integrated health surveillance model [33,34]. Moreover, the identification of symptoms and comorbidities that are predictors of severe illness and intensive care unit (ICU) admission in these populations can help in the risk stratification of each patient in health services and facilitate more effective planning and mobilization of resources [35,36,37].

This study has the following objectives: (1) to investigate the relationship between SDH, incidence, and mortality; (2) to verify which sociodemographic factors, symptoms, and comorbidities predict clinical management; and (3) to analyze which sociodemographic and clinical factors are associated with lower survival of individuals with COVID-19 in the Xingu Health Region in the eastern Brazilian Amazon. This study hypothesized that poorer indicators of SDH are related to higher rates of incidence and mortality and that variables including advanced age, the female sex, living in cities with poorer social development, and specific symptoms and comorbidities serve as predictors of more severe clinical management, such as medical ward and intensive care, and lower survival in individuals with COVID-19 in the health region of Xingu in the eastern Brazilian Amazon.

## 2. Materials and Methods

### 2.1. Study Design

An observational study design employing an ecological approach with descriptive and inferential analyses was adopted.

### 2.2. Study Population and Period

This study examined the secondary data of individuals diagnosed with COVID-19 between March 2020 and March 2021 in a pre-vaccination scenario in the Xingu Health Region, Pará State, Brazil.

### 2.3. Inclusion and Exclusion Criteria

Diagnosis of SARS-CoV-2 by a reverse transcription-polymerase chain reaction (RT-PCR) or a serological test (rapid test) was defined as the study inclusion criterion. Therefore, only cases that were RT-PCR- or rapid test-positive were included in the study. Cases with missing data (i.e., sociodemographic information) were excluded.

### 2.4. Setting

The data obtained from the nine municipalities of the Xingu Health Region, Pará State, Brazil (Figure 1), were analyzed. To visualize these cities, a map was constructed using QGIS Desktop 3.26.

### 2.5. Assessments

The databases of SESPA, the Brazilian Institute of Geography and Statistics (IBGE), e-Gestor AB, and the National Institute for Space Research (INPE) were utilized. The open-access database maintained by SESPA includes information regarding cases identified as COVID-19-positive, reported daily by municipal health departments and health services as part of the COVID-19 surveillance system in the state of Pará. These data are published daily between 5:00 and 7:00 p.m. on the www.covid-19.pa.gov.br portal; this portal comprises all the data on the COVID-19 pandemic in Pará [36]. The IBGE database includes data on Brazil, its states, and its municipalities; it includes infographics, maps, and other information on topics such as education, labor, the economy, population, health, and territory [21,37]. The e-Gestor AB database [38,39] is a platform that provides access to primary healthcare (PHC) information systems for the management of PHC data by managers and health professionals, facilitating access to data that can be useful in the organization and planning of health services. The INPE database includes environmental information on activities conducted by top research institutes in the country following minimum quality standards to facilitate the understanding and reuse of information [40].

The sociodemographic factors analyzed included age, sex, and municipality of residence. The clinical symptoms evaluated included fever, cough, dyspnea, nausea, headache, runny nose, nasal congestion, sore throat, diarrhea, chills, conjunctivitis, odynophagia, anosmia, ageusia, adynamia, myalgia, and arthralgia. The comorbidities considered included obesity; asthma; diabetes; immunodeficiency diseases; heart disease; pneumopathy; and neurological, renal, hematological, and hepatic diseases. The clinical management analysis included home care and the need for hospitalization, separating those who needed a medical ward from those who were hospitalized in intensive care units (ICUs). The time to symptom onset, date of death, and mortality of individuals were recorded. These data were obtained from the database provided by SESPA [36]. The incidence and mortality rates were calculated subsequently. IBGE [21,37] data were used to estimate the number of individuals in the Brazilian population (2020) in terms of the calculation of incidence and mortality rates, overall and according to sex, per 1000 inhabitants.

The SDH indicators used in the analysis of the Xingu Health Region are provided in Table 1. The SDH indicators analyzed included sociodemographic and habitation factors (population density [21,37]; percentage of elderly in the population [21,37]; percentage of the vulnerable households with an older adult in the population [21,37]; percentage of people in households with walls not made of masonry or wood [21,37]), economic and environmental factors (Gini index [21,37]; gross domestic product (GDP) [21,37]; Human Development Index [21,37]; percentage urbanization of public roads [41]; hotspot concentration [40]), health and resources (percentage of primary health care coverage [38,39]; percentage of people with health insurance [38,39]; number of physicians per 1000 individuals [38,39]; public health expenditure in the municipality, in BRL/inhabitant, per capita), and education and work (schooling sub-index [21,39]; illiteracy rate at age 15 and above [21,37]; unemployment rate at 10 years or older [21,37]). Classification of the indicators was based on the social gradients in health as theorized by Dalgreen and Whitehead [42] with respect to SDH [30].

### 2.6. Statistical Analysis

Descriptive statistical analysis was conducted to compute frequencies (absolute and relative), means, and standard deviations (parametric) or medians with interquartile range (IQR, non-parametric) for each group. The incidence and mortality rates for every 1000 individuals were also calculated. For the spatial representation of incidence and mortality in the Xingu Health Region, the values of the four classes were constructed based on the equal interval technique, which is based on the amplitude of the data.

Bivariate correlation coefficients, Pearson’s r (parametric), and Spearman’s rs (nonparametric), were used to verify the level of correlation between variables. Binary logistic regression analysis was used to establish the determining factors for clinical management and mortality. Initially, univariate analysis was performed considering a *p*-value of <0.25. To verify multicollinearity, the variance inflation factor (VIF) was calculated, and variables that presented a VIF value above 10 were removed from the final model. Statistical significance was set at *p* < 0.05. An odds ratio (OR) with a 95% confidence interval (95% CI) was used to quantify the degree of association.

Survival curves were obtained by using the Kaplan–Meier estimator; additionally, log-rank (initial), Breslow (intermediary), and Tarone–Ware (final) tests were used to identify statistically significant differences in the different periods [42].

SPSS Version 26.0 (IBM Corp. Released 2019. IBM SPSS Statistics for Windows, Version 26.0. IBM Corp., Armonk, NY, USA) was used for the statistical analyses.

### 2.7. Ethical Issues

The open-access database used in this study is maintained by the State of Pará Public Health Secretary (SESPA), a state government agency, and contains consolidated information pertaining to individuals who have sought healthcare services owing to COVID-19-related symptoms. The individuals covered in this database are not identified; hence, according to National Health Council (CNS) Resolution No. 510 of 7 April 2016, evaluation by the relevant research ethics committee was not required [43].

## 3. Results

Overall, 20,296 COVID-19 cases were identified during the study period. However, 117 cases were excluded due to the incomplete sociodemographic information of these patients. Furthermore, 4041 cases were excluded since no other information except the clinical diagnosis was available. The final sample included 16,138 patients (Figure 2).

The incidence rate of COVID-19 per 1000 inhabitants in the Xingu Health Region was 45.59 and the mortality rate was 1.01. In descending order, the highest incidence rate was observed in Vitória do Xingu (90.84), followed by Altamira (56.83), Senador José Porfírio (56.62), Brasil Novo (54.19), Medicilândia (48.07), Anapu (39.75), Pacajá (33.07), Porto de Moz (28.66), and Uruará (26.94). However, most cases were reported in Altamira (41%), Pacajá (9.9%), and Medicilândia (9.5%). It was observed that 359 individuals did not survive being infected with COVID-19. In descending order, Altamira (1.62) showed the highest mortality rate, followed by Brasil Novo (1.47), Vitória do Xingu (1.44), Senador José Porfírio (1.31), Anapú (0.87), Porto de Moz (0.67), Medicilândia (0.63), Uruará (0.42), and Pacajá (0.41). Deaths were concentrated in Altamira (52%), Porto de Moz (7.8%), and Anapú (7%) (Figure 3).

### 3.1. Relationship among the SDH Indicators, Incidence, and Mortality

The incidence and mortality rates according to sex and SDH indicators in all municipalities in the Xingu Health Region are presented in Table 1.

When assessing the correlation of incidence and SDH indicators, correlations with the following variables were observed: GDP (rs = 0.8000, *p* = 0.013), percentage of people with health insurance (rs = 0.8000, *p* = 0.013), public health expenditure in the municipality (rs = 0.7667, *p* = 0.021), and percentage of the population residing in households with walls not made of masonry or wood (r = −0.6764, *p* = 0.040).

When assessing the correlation of mortality and SDH indicators, correlations with the following variables were observed: percentage of people with health insurance (rs = 0.8333, *p* = 0.008), public health expenditure in the municipality (rs = 0.7333, *p* = 0.030), and percentage of the population residing in households with walls not made of masonry or wood (r = −0.6812, *p* = 0.040). The other SDH indicators were not significant for incidence or mortality.

### 3.2. Sociodemographic Factors, Symptoms and Comorbidities as Predictors of Clinical Management

The sociodemographic factors associated with admission to a medical ward included age and residing in Brasil Novo. However, the female sex, residing in Uruará and Vitória do Xingu, and being locally diagnosed were associated with a lower chance of being admitted to a medical ward. Regarding symptoms, fever, cough, emesis, and dyspnea were associated with admission to a medical ward. Individuals with a headache, sore throat, myalgia, and arthralgia were less likely to be admitted to a medical ward. The predictive comorbidities for admission to a medical ward included diabetes, heart disease, neurological diseases, kidney diseases, and obesity (Table 2).

The sociodemographic factors including advanced age and having an imported case were associated with admission to an ICU. However, the female sex and residing in the municipalities of Brasil Novo, Pacajá, and Vitória do Xingu, compared with Altamira, were protective factors for ICU admission (Table 3).

Emesis, chills, and dyspnea were predictors of ICU admission. However, patients with a headache and sore throat were less likely to be hospitalized. The comorbidities of diabetes, heart disease, and obesity were identified as risk factors for ICU admission (Table 3).

The predictors for admission to ICU were having an imported case, chills, dyspnea, and diabetes. The female sex and residing in the municipalities of Brasil Novo, Medicilândia, Pacajá, and Vitória do Xingu, in comparison with Altamira, were considered protective factors for ICU admission (Table 4).

### 3.3. Sociodemographic Factors, Symptoms and Comorbidities as Predictors of Mortality

A total of 359 individuals (2.2 %) died due to being infected with COVID-19 during the study period. The sociodemographic factors predicting higher mortality included advanced age and residing in the municipality of Anapu. Having an imported case and residing in Pacajá were correlated with lower mortality rates. Individuals with dyspnea and fever had a higher risk of death. However, individuals with a sore throat and anosmia showed less association. Regarding comorbidities, heart disease was correlated with a higher chance of death. According to clinical management, patients admitted to ICU were most likely to die, followed by those admitted to a medical ward. Other variables did not present significant differences (Table 5).

### 3.4. Survival Analysis

Only the variable age group showed an association with COVID-19 survival, and individuals aged 60 years or older exhibited the lowest survival rate. The results of the Kaplan–Meier survival analysis conducted for COVID-19, according to sex, age group, home city, symptom, and commodity, are presented in Figure 4.

## 4. Discussion

Amazonian communities continue to face particular challenges in relation to the COVID-19 pandemic, owing to the fact that each community has a different type of organizational structure and a different type of social and cultural behavior [44]. Moreover, in Brazil, the denialist administration of the federal government, which has operated without a unified policy to combat and control the COVID-19 disease, has reinforced historical structural inequalities and regional vulnerabilities [45]. This has adversely affected vulnerable populations in rural and remote areas as well as traditional peoples (indigenous peoples, forest peoples, quilombolas, and riverine peoples) who reside in the Brazilian Amazon [46]. Additionally, it must be noted that socioeconomic inequalities and limited access to health services can contribute to increased incidence and mortality [47].

The COVID-19 incidence rate distribution per 1000 inhabitants varied in each municipality of the Xingu Health Region, with the highest incidence rate in Vitória do Xingu (90.84) and the lowest incidence rate in Uruará (26.94). Vitória do Xingu is a port city, through which a significant number of individuals move to other municipalities of the Xingu region due to its connection with the Transamazonian region, as well as with the Amazon River, which differs from Uruará. This may explain the high incidence in this city. Port areas are common epicenters of disease transmission, as demonstrated by the outbreak of the Spanish influenza epidemic in the city of Recife, in northeast Brazil, in the early 20th century; the influenza virus in this case was transmitted via a British ship docked in the city port [48].

Regarding COVID-19 mortality, our study showed that 40.8% of the confirmed COVID-19 cases and 52% of deaths were reported in Altamira, in addition to residence in Altamira being a risk factor for ICU admission. Thus, the confirmed cases were distributed more between cities than mortality. The dependency illustrated the relationship between some Brazilian northern municipalities and highlighted how the virus spread intensely in a local city due to the influence of two other bigger cities [49]. Altamira is the urban center and most populous city of the Xingu Health Region, where the majority of healthcare professionals and specialized services are concentrated. Therefore, it can be supposed that for moderate or severe cases (more likely to evolve to death), these individuals sought healthcare in Altamira. Furthermore, these findings indicate that patients did not stay isolated in their houses but continued to travel between municipalities, especially due to the region’s economic characteristics, of which the export production chain occupies a significant percentage.

Vitória do Xingu and Altamira were the cities with the highest number of foreign cases and highest incidence rates. Similar findings were obtained for mortality, with the highest rates observed in Altamira, Brasil Novo, and Vitória do Xingu. The geography of the Xingu region could explain these findings. As can be seen in Figure 1, both cities are located on the Xingu River, and the existing ports in these cities simultaneously ship a variety of products and people. Despite this economic infrastructure and connection between these cities, wealth is not well distributed [50], and the COVID-19 pandemic started in cities with better socioeconomic conditions and subsequently migrated to more vulnerable local communities [51,52].

Confirming these results, positive correlations between incidence and GDP, the percentage of people with health insurance, and public health expenditure in the municipality were observed in this study. Additionally, a negative correlation was observed between incidence and the percentage of the population residing in households with walls not made of masonry or wood. A positive correlation was observed between the percentage of people with health insurance and public health spending in the municipality. Further, a negative correlation was observed between mortality and the percentage of the population residing in households with walls not made of masonry or wood. Healthcare workers, ICU beds, and mechanical ventilators, which are frequently needed in COVID-19 severe cases, are unequally distributed in Brazil [53,54]. Considering the above, it can be assumed that the spread of COVID-19 and its associated mortality were influenced by high SDH indicators in the economic, health, and resource sectors, as well as by interdependence between cities. Additionally, political and socioeconomic factors were critical to the spatial and temporal dynamics of COVID-19 outcomes in Brazil, especially in the first wave, in which the largest municipalities with a higher socioeconomic profile were the most affected [53]. It was also observed that the municipalities with better coverage indicators and health resources had higher incidence and mortality rates, suggesting misuse of these resources [54].

Considering sociodemographic characteristics, sex-disaggregated mortality and morbidity surveillance data should be a priority in COVID-19 research [55]. Sex differences in viral transmission and disease progression deserve explicit attention due to different levels of exposure between men and women; as comorbidities are usually more prevalent in men, this can be linked with evidence indicating that males are more associated with severely affected with COVID-19 and death [56]. It has also been suggested that the exposure of women to the COVID-19 virus may be higher than that of men, since frontline providers are generally women, comprising 70% of the global health and social care workforce [57]. This study’s results show that females formed the highest proportion in the group that stayed in home care and recovered, were less associated with the need for hospitalization, and were less likely to die. It is not yet well investigated whether biological differences [57] and lifestyle habits, or a combination of these, are the main factors associated with death due to COVID-19 among males.

We also found that older age was associated with the need for hospitalization and death. This can be attributed to the fact that the aging process leads to several changes that increase COVID-19 susceptibility, such as immunosenescence, changes in T-cell diversity, inflammation, a dysregulated renin-angiotensin system, changes in the glycome, advanced biological age, and epigenetic changes [58], that impair the autoimmunity of the individual in addition to the presence of comorbidities [59].

Despite the variety of symptoms investigated, fever, cough, and dyspnea were the most prevalent among patients who required hospitalization; this result is similar to other studies that evaluated symptomatic patients [60,61,62]. We observed that individuals with these symptoms and emesis were more likely to be hospitalized in a medical ward (moderate cases). Regarding ICU admission (severe cases), emesis, dyspnea, and chills remained as risk factors. Systematic and non-systematic reviews have reported that fever, dyspnea, chills, and gastrointestinal symptoms were associated with severe COVID-19 infection and ICU admission [63,64,65,66].

Additionally, headache, sore throat, and myalgia/arthralgia were less associated with the need for hospitalization and were consequently related to a better prognosis. In a systematic review and meta-analysis that investigated predictors and outcomes, an association between headache, myalgia/arthralgia, and COVID-19 severity was not observed [67], even though these were prevalent in many studies. This study shows that these symptoms were more prevalent in homecare cases and did not vary between those that needed hospitalization; therefore, it can be supposed that despite being cited in many studies, these symptoms did not present a risk factor in relation to illness severity.

Diabetes, heart disease, and obesity were consistently associated with the need for hospitalization, either in a medical ward or ICU. The existing literature provides sufficient evidence to support the role of each comorbidity as a risk factor for severe diseases [12,65,66,67,68]. Low-grade chronic inflammation, a compromised immune response, and prothrombotic status are the complications associated with diabetes and obesity [59]. In relation to heart disease, one of the phases of viral action is the inhibition of the Angiotensin 2-Converter enzyme (ECA-2), which could deregulate the renin-angiotensin-aldosterone system, causing local and systemic tissue lesions [68]. Additionally, kidney and neurological diseases were associated with the need for hospitalization in a medical ward. However, there have been limited studies on the association between preexisting kidney diseases and COVID-19, and most studies in the literature have shown that kidney-related conditions are associated with the risk of mortality [60,62,63,66,67,68]. In relation to neurological diseases, a systematic review has pointed out that mental and neurological disorders were associated with COVID-19 disease severity and further with mortality [69].

The literature provides differing evidence with respect to symptoms as predictors of death. However, this study found that individuals with fever or dyspnea were more likely to die. This finding is in line with that of a Nigerian retrospective cohort study [70]. However, systematic reviews have not observed any symptoms to be a factor associated with COVID-19 mortality [63,68]. Regarding baseline characteristics, such as age and the presence of comorbidities, this study found that older individuals with heart disease were more likely to die, and this is well-evidenced in the existing scientific literature [59,68].

By identifying the symptoms and comorbidities that are predictive of serious illness and ICU admission, risk stratification within health services becomes possible. This can lead to management changes and improvements in resource allocation efficiency for patients at a higher risk of serious infection from COVID-19. Such identification would also facilitate more informed discussions regarding the predicted clinical trajectory, allowing for more accurate and timely advanced care planning. Similarly, it would assist the public health response mechanism in controlling the spread of the disease since knowledge about the different prevalence and risks of various conditions can help to focus and adapt public health efforts [71].

The limitations of the present study include the limitations that accompany the process of conducting an ecological study: errors may arise in the filling out of patient data; the underreporting of cases in terms of race or ethnicity, which was not reported in the SESPA database; and a large amount of unknown or unreported data, which affects the reliability of the analysis. Nonetheless, despite these issues, the situation has improved in recent years in Brazil [72]. Another limitation of the ecological design is the acquisition of data on SDH indicators since these data are collected from different sources and different periods [21,36,37,38,39,40]. Although ecological research makes causal inferences about individuals based on group observations, this study can contribute to the evaluation of public health policies in the communities of the Xingu Health Region, especially in the area of clinical management and surveillance of COVID-19, and provide avenues for future studies using other methodological approaches.

## 5. Conclusions

Incidence and mortality were higher in the cities of Vitória do Xingu and Altamira. Additionally, it was found that a higher percentage of people with health insurance and higher public health expenditure in the municipality was associated with higher incidence and mortality. A higher GDP was associated with a higher incidence. The female sex was associated with better clinical management. Residence in Altamira was a risk factor for ICU admission. Symptoms and comorbidities that predicted worse clinical management included dyspnea, fever, emesis, chills, diabetes, cardiac and renal diseases, obesity, and neurological diseases. Headache, anosmia, sore throat, myalgia, and arthralgia were protective factors. With advancing age, patients with COVID-19 face a higher incidence, mortality, and lower survival rates.

Therefore, it can be concluded that SDH indicators, symptomatology, and comorbidities have direct implications for the incidence, mortality, and clinical management of COVID-19 in the Xingu Health Region in eastern Amazonia, Brazil.

## Figures and Tables

**Figure 1 ijerph-20-04639-f001:**
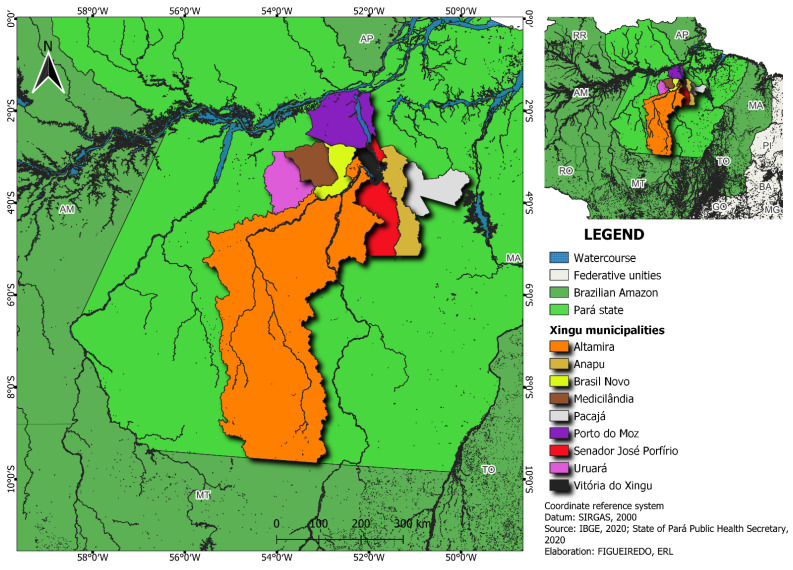
Map of the Xingu Health Region.

**Figure 2 ijerph-20-04639-f002:**
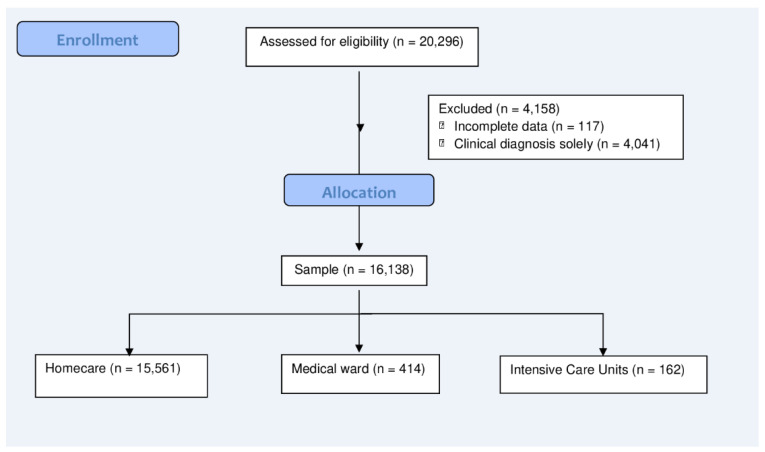
Flow diagram of the selection and distribution of individuals in the groups.

**Figure 3 ijerph-20-04639-f003:**
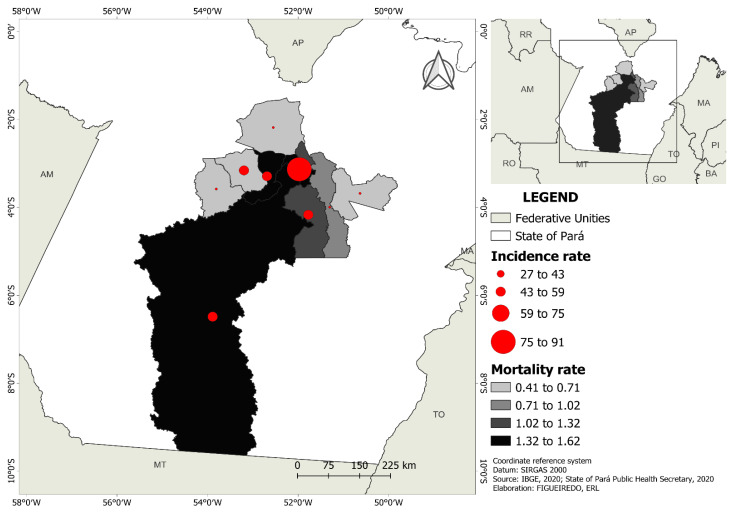
Incidence and mortality rate distribution per 1000 inhabitants of municipalities in the Xingu Health Region.

**Figure 4 ijerph-20-04639-f004:**
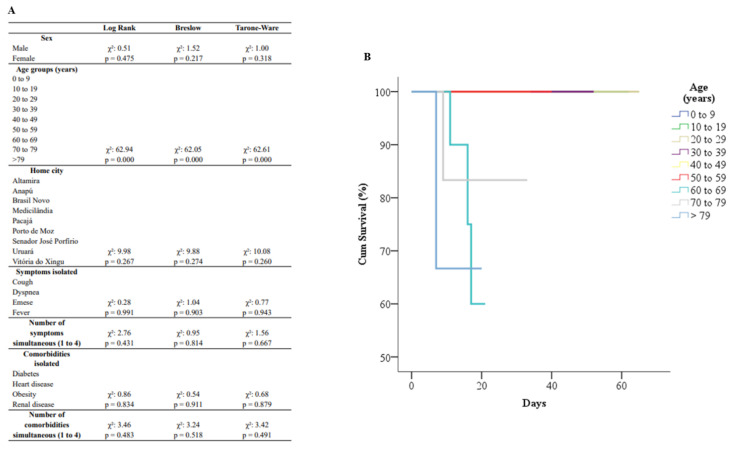
Kaplan–Meier survival analysis for COVID-19 by sex, age groups, home city, symptoms and comorbidities (**A**); representation graphic for age groups (**B**).

**Table 1 ijerph-20-04639-t001:** Exploratory analysis of SDH indicators in the study of incidence and mortality by COVID-19 in the Xingu Health Region.

Social Determinants of Health	Xingu Health Region	Altamira	Anapu	Brasil Novo	Medicilândia	Pacajá	Porto de Moz	Senador José Porfírio	Uruará	Vitória do Xingu
**Sex**										
Incidence in females	55.42	62.09	48.40	65.43	59.97	44.14	40.31	69.88	35.31	117.77
Incidence in males	44.37	60.43	35.45	51.55	46.79	29.98	26.43	48.38	25.65	78.61
Mortality in females	1.38	2.10	1.17	1.22	0.54	0.45	1.00	1.66	0.78	2.59
Mortality in males	0.71	1.13	0.64	1.23	0.48	0.32	0.51	0.97	0.23	0.73
**Sociodemographic and housing**										
Population density (2010)	2.81	0.71	2.28	2.39	3.77	3.97	2.32	0.82	4.22	4.85
Percentage of elderly in population (2020)	8.58	10.01	7.35	13.23	8.91	8.32	5.00	8.05	7.97	8.41
Percentage in vulnerable households with the elderly in the population (2010)	6.15	4.4	5.06	5.90	3.93	6.12	7.50	10.40	4.72	7.40
Percentage of the population residing in households with walls not made of masonry or wood (2010)	18.15	12.24	29.51	15.38	20.17	27.84	18.31	13.95	17.33	8.66
**Economic and environmental factors**										
Gini index (2010)	0.6084	0.572	0.585	0.684	0.614	0.663	0.599	0.566	0.586	0.606
Gross Domestic Product in BRL (GDP) (2020)	43,941.75	24,250.13	14,926.79	18,362.96	20,744.58	14,223.20	7842.34	14,017.34	13,431.86	267,676.58
Human Development Index (HDI) (2010)	0.569	0.665	0.548	0.613	0.582	0.515	0.503	0.514	0.589	0.596
Percentage urbanization of public roads (2010)	4.79	22.70	0.40	2.40	0.70	0.40	4.40	6.30	3.43	2.40
Hotspot concentration (2017)	5.25	23.62	3.39	1.78	2.89	6.39	2.25	2.99	3.43	0.51
**Health and resources**										
Primary health care coverage in % (2020)	66.85	63.22	74.22	100	65.51	43.39	58.71	29.59	75.86	91.19
Percentage of people with health insurance (2017)	1.21	4.72	0.9	0.84	0.47	0.24	0.25	1.3	0.64	1.58
Number of physicians per 1000 individuals (2021)	0.41	1.22	0.28	0.26	0.22	0.18	0.19	0.42	0.30	0.66
Public health expenditure in the municipality, in BRL/inhabitant, per capita (2021)	887.99	731.29	657.99	1120.96	597.16	602.70	549.97	841.30	624.57	2266.03
**Education and work**										
Schooling sub-index (2010)	0.294	0.461	0.278	0.293	0.266	0.246	0.248	0.207	0.342	0.307
Illiteracy rate at age 15 and above (2010)	18.11	12.45	19.24	16.09	18.35	21.53	21.46	22.71	15.11	16.09
Unemployment rate at 10 years or older (2010)	5.44	6.96	3.51	4.06	2.72	4.81	6.93	5.76	6.56	7.71

**Table 2 ijerph-20-04639-t002:** Sociodemographic factors, symptoms and comorbidities as predictors of admission to a medical ward in relation to home care.

Characteristic	Home Care N = 15,561 ^1^	Medical Ward N = 414 ^1^	N	OR ^2^	95% CI ^2^	*p*-Value	aOR ^3^	95% CI ^3^	*p*-Value
Age	38 (27, 50)	64 (46, 77)	15,975	1.06	1.06, 1.07	<0.001	1.05	1.04, 1.05	<0.001
Municipality			15,975			<0.001			
Altamira	6297 (40%)	194 (47%)		—	—		—	—	
Anapu	1112 (7.1%)	18 (4.3%)		0.53	0.31, 0.83		0.61	0.34, 1.02	0.072
Brasil Novo	769 (4.9%)	38 (9.2%)		1.60	1.11, 2.26		2.02	1.30, 3.07	0.001
Medicilândia	1505 (9.7%)	28 (6.8%)		0.60	0.40, 0.89		0.77	0.46, 1.23	0.29
Pacajá	1551 (10.0%)	38 (9.2%)		0.80	0.55, 1.12		0.70	0.45, 1.05	0.092
Porto de Moz	1138 (7.3%)	44 (11%)		1.25	0.89, 1.73		1.15	0.74, 1.73	0.53
Senador José Porfírio	622 (4.0%)	22 (5.3%)		1.15	0.71, 1.76		1.19	0.69, 1.96	0.51
Uruará	1205 (7.7%)	11 (2.7%)		0.30	0.15, 0.52		0.42	0.20, 0.79	0.012
Vitória do Xingu	1362 (8.8%)	21 (5.1%)		0.50	0.31, 0.77		0.35	0.19, 0.61	<0.001
Sex			15,975						
Male	7208 (46%)	254 (61%)		—	—		—	—	
Female	8353 (54%)	160 (39%)		0.54	0.44, 0.66	<0.001	0.68	0.54, 0.86	0.001
Local case	14,815 (95%)	377 (91%)		0.51	0.37, 0.74	<0.001	0.41	0.25, 0.72	0.001
Imported case	221 (1.4%)	21 (5.1%)	15,975	3.71	2.28, 5.73	<0.001	1.47	0.72, 3.00	0.29
Fever	9532 (61%)	293 (71%)	15,975	1.53	1.24, 1.90	<0.001	1.34	1.03, 1.76	0.031
Cough	8724 (56%)	315 (76%)	15,975	2.49	1.99, 3.15	<0.001	1.87	1.42, 2.47	<0.001
Emesis	463 (3.0%)	32 (7.7%)	15,975	2.73	1.85, 3.90	<0.001	3.55	2.17, 5.60	<0.001
Headache	5637 (36%)	50 (12%)	15,975	0.24	0.18, 0.32	<0.001	0.48	0.34, 0.67	<0.001
Runny nose	1548 (9.9%)	10 (2.4%)	15,975	0.22	0.11, 0.40	<0.001	0.54	0.25, 1.04	0.089
Nasal congestion	246 (1.6%)	6 (1.4%)	15,975	0.92	0.36, 1.89	0.830			
Sore throat	6758 (43%)	112 (27%)	15,975	0.48	0.39, 0.60	<0.001	0.38	0.29, 0.49	<0.001
Myalgia and arthralgia	3642 (23%)	38 (9.2%)	15,975	0.33	0.23, 0.46	<0.001	0.57	0.38, 0.82	0.003
Diarrhea	1360 (8.7%)	38 (9.2%)	15,975	1.06	0.74, 1.46	0.757			
Chills	193 (1.2%)	1 (0.2%)	15,975	0.19	0.01, 0.86	0.026	0.35	0.02, 1.83	0.33
Adynamia	584 (3.8%)	20 (4.8%)	15,975	1.30	0.80, 2.00	0.275			
Odynophagy	70 (0.4%)	0 (0%)	15,975	0.00	0.00, 0.09	0.055			
Anosmia	3166 (21%)	33 (9.1%)	15,517	0.38	0.26, 0.53	<0.001	1.00	0.54, 1.81	>0.99
Ageusia	2950 (19%)	29 (8.0%)	15,517	0.36	0.24, 0.52	<0.001	0.62	0.32, 1.16	0.14
Dyspnea	4097 (26%)	290 (70%)	15,975	6.54	5.30, 8.12	<0.001	4.01	3.14, 5.14	<0.001
Conjunctivitis	27 (0.2%)	0 (0%)	15,975	0.00	0.00, 13.0	0.234	0.00	0.00, 17.0	0.97
Diabetes	272 (1.7%)	67 (16%)	15,975	10.9	8.09, 14.4	<0.001	3.15	2.13, 4.61	<0.001
Immunodeficiency	116 (0.7%)	4 (1.0%)	15,975	1.30	0.40, 3.11	0.623			
Heart diseases	399 (2.6%)	73 (18%)	15,975	8.13	6.16, 10.6	<0.001	2.16	1.45, 3.19	<0.001
Lung diseases	65 (0.4%)	1 (0.2%)	15,975	0.58	0.03, 2.62	0.550			
Neurological diseases	1 (<0.1%)	1 (0.2%)	15,975	37.7	1.49, 954	0.032	168	6.14, 4,658	<0.001
Kidney diseases	24 (0.2%)	9 (2.2%)	15,975	14.4	6.29, 30.1	<0.001	5.45	1.80, 14.9	0.002
Obesity	12 (<0.1%)	12 (2.9%)	15,975	38.7	17.1, 87.5	<0.001	13.9	4.60, 42.3	<0.001
Asthma	30 (0.2%)	2 (0.5%)	15,975	2.51	0.41, 8.35	0.268			
Blood diseases	3 (<0.1%)	2 (0.5%)	15,975	25.2	3.31, 152	0.005	22.6	1.01, 515	0.059
Liver diseases	3 (<0.1%)	2 (0.5%)	15,975	25.2	3.31, 152	0.005	22.6	1.01, 515	0.059

^1^ Median (IQR); n (%); ^2^ OR = Odds Ratio, CI = Confidence Interval; ^3^ aOR = Adjusted Odds Ratio.

**Table 3 ijerph-20-04639-t003:** Sociodemographic factors, symptoms, and comorbidities as predictors of admission to intensive care units (ICU) compared with home care.

Characteristic	Home Care N = 15,561 ^1^	ICU N = 162 ^1^	N	OR ^2^	95% CI ^2^	*p*-Value	aOR ^3^	95% CI ^3^	*p*-Value
**Age**	38 (27, 50)	66 (50, 75)	15,723	1.07	1.06, 1.08	<0.001	1.04	1.03, 1.06	<0.001
**Municipality**			15,723			<0.001			
Altamira	6297 (40%)	100 (62%)		—	—		—	—	
Anapu	1112 (7.1%)	7 (4.3%)		0.40	0.17, 0.79		0.45	0.17, 0.99	0.069
Brasil Novo	769 (4.9%)	5 (3.1%)		0.41	0.14, 0.91		0.26	0.07, 0.73	0.021
Medicilândia	1505 (9.7%)	4 (2.5%)		0.17	0.05, 0.40		0.05	0.01, 0.17	<0.001
Pacajá	1551 (10.0%)	11 (6.8%)		0.45	0.23, 0.80		0.36	0.15, 0.74	0.011
Porto de Moz	1138 (7.3%)	16 (9.9%)		0.89	0.50, 1.46		0.71	0.34, 1.37	0.33
Senador José Porfírio	622 (4.0%)	6 (3.7%)		0.61	0.24, 1.28		0.29	0.06, 0.91	0.068
Uruará	1205 (7.7%)	8 (4.9%)		0.42	0.19, 0.81		0.44	0.17, 0.98	0.060
Vitória do Xingu	1362 (8.8%)	5 (3.1%)		0.23	0.08, 0.51		0.11	0.02, 0.34	<0.001
**Sex**			15,723			<0.001			
Male	7208 (46%)	110 (68%)		—	—		—	—	
Female	8353 (54%)	52 (32%)		0.41	0.29, 0.56		0.45	0.30, 0.66	<0.001
**Local case**	14,815 (95%)	144 (89%)	15,723	0.40	0.25, 0.68	0.001	1.26	0.42, 4.72	0.71
**Imported case**	221 (1.4%)	21 (13%)	15,723	10.3	6.25, 16.3	<0.001	6.79	1.94, 26.0	0.004
**Fever**	9532 (61%)	115 (71%)	15,723	1.55	1.11, 2.20	0.010	1.22	0.80, 1.90	0.37
**Cough**	8724 (56%)	116 (72%)	15,723	1.98	1.41, 2.81	<0.001	1.32	0.85, 2.07	0.23
**Emesis**	463 (3.0%)	16 (9.9%)	15,723	3.57	2.04, 5.85	<0.001	6.00	2.91, 11.6	<0.001
**Headache**	5637 (36%)	16 (9.9%)	15,723	0.19	0.11, 0.31	<0.001	0.45	0.24, 0.81	0.011
**Runny nose**	1548 (9.9%)	5 (3.1%)	15,723	0.29	0.10, 0.63	<0.001	1.00	0.32, 2.59	>0.99
**Nasal congestion**	246 (1.6%)	2 (1.2%)	15,723	0.78	0.13, 2.45	0.715			
**Sore throat**	6758 (43%)	45 (28%)	15,723	0.50	0.35, 0.70	<0.001	0.48	0.31, 0.74	0.001
**Myalgia and arthralgia**	3642 (23%)	15 (9.3%)	15,723	0.33	0.19, 0.55	<0.001	0.71	0.38, 1.27	0.28
**Diarrhea**	1360 (8.7%)	13 (8.0%)	15,723	0.91	0.49, 1.55	0.745			
**Chills**	193 (1.2%)	4 (2.5%)	15,723	2.02	0.62, 4.83	0.216	5.14	1.14, 16.2	0.013
**Adynamia**	584 (3.8%)	6 (3.7%)	15,723	0.99	0.39, 2.05	0.974			
**Odynophagy**	70 (0.4%)	0 (0%)	15,723	0.00	0.00, 0.23	0.228	0.00	0.00, 0.00	0.98
**Anosmia**	3166 (21%)	5 (3.4%)	15,303	0.13	0.05, 0.29	<0.001	0.56	0.13, 1.90	0.39
**Ageusia**	2950 (19%)	5 (3.4%)	15,303	0.14	0.05, 0.32	<0.001	0.43	0.11, 1.46	0.21
**Dyspnea**	4097 (26%)	136 (84%)	15,723	14.6	9.78, 22.8	<0.001	10.2	6.36, 17.0	<0.001
**Conjunctivitis**	27 (0.2%)	0 (0%)	15,723	0.00	0.00, 33.0	0.454			
**Diabetes**	272 (1.7%)	42 (26%)	15,723	19.7	13.4, 28.3	<0.001	4.92	2.91, 8.18	<0.001
**Immunodeficiency**	116 (0.7%)	5 (3.1%)	15,723	4.24	1.48, 9.51	0.010	0.71	0.21, 1.96	0.55
**Heart diseases**	399 (2.6%)	42 (26%)	15,723	13.3	9.14, 19.0	<0.001	3.94	2.32, 6.58	<0.001
**Lung diseases**	65 (0.4%)	3 (1.9%)	15,723	4.50	1.09, 12.3	0.039	2.42	0.48, 8.59	0.22
**Neurological diseases**	1 (<0.1%)	2 (1.2%)	15,723	194	18.5, 4,198	<0.001	22.9	0.11, 3,655	0.37
**Kidney diseases**	24 (0.2%)	4 (2.5%)	15,723	16.4	4.78, 43.0	<0.001	1.32	0.14, 7.91	0.78
**Obesity**	12 (<0.1%)	9 (5.6%)	15,723	76.2	30.7, 183	<0.001	48.0	14.5, 167	<0.001
**Asthma**	30 (0.2%)	0 (0%)	15,723	0.00	0.00, 9.15	0.430			
**Blood diseases**	3 (<0.1%)	0 (0%)	15,723	0.00		0.803			
**Liver diseases**	3 (<0.1%)	0 (0%)	15,723	0.00		0.803			

^1^ Median (IQR); n (%); ^2^ OR = Odds Ratio, CI = Confidence Interval; ^3^ aOR = Adjusted Odds Ratio.

**Table 4 ijerph-20-04639-t004:** Sociodemographic factors, symptoms, and comorbidities as predictors of admission to intensive care units (ICU) in relation to admission to a medical ward.

Characteristic	Medical Ward N = 414 ^1^	ICU N = 162 ^1^	N	OR ^2^	95% CI ^2^	*p*-Value	aOR ^3^	95% CI ^3^	*p*-Value
**Age**	64 (46, 77)	66 (50, 75)	576	1.00	1.00, 1.01	0.378			
**Municipality**			576			0.008			
Altamira	194 (47%)	100 (62%)		—	—		—	—	
Anapu	18 (4.3%)	7 (4.3%)		0.75	0.29, 1.79		0.48	0.15, 1.32	0.18
Brasil Novo	38 (9.2%)	5 (3.1%)		0.26	0.09, 0.61		0.18	0.05, 0.50	0.002
Medicilândia	28 (6.8%)	4 (2.5%)		0.28	0.08, 0.73		0.13	0.03, 0.44	0.003
Pacajá	38 (9.2%)	11 (6.8%)		0.56	0.26, 1.11		0.28	0.10, 0.67	0.007
Porto de Moz	44 (11%)	16 (9.9%)		0.71	0.37, 1.29		0.56	0.26, 1.15	0.13
Senador José Porfírio	22 (5.3%)	6 (3.7%)		0.53	0.19, 1.27		0.28	0.06, 0.91	0.056
Uruará	11 (2.7%)	8 (4.9%)		1.41	0.53, 3.60		1.83	0.65, 5.03	0.24
Vitória do Xingu	21 (5.1%)	5 (3.1%)		0.46	0.15, 1.17		0.24	0.05, 0.82	0.038
**Sex**			576			0.140			
Male	254 (61%)	110 (68%)		—	—		—	—	
Female	160 (39%)	52 (32%)		0.75	0.51, 1.10		0.59	0.37, 0.92	0.021
**Local case**	377 (91%)	144 (89%)	576	0.79	0.44, 1.45	0.431			
**Imported case**	21 (5.1%)	21 (13%)	576	2.79	1.47, 5.28	0.002	3.00	1.32, 6.85	0.009
**Fever**	293 (71%)	115 (71%)	576	1.01	0.68, 1.52	0.959			
**Cough**	315 (76%)	116 (72%)	576	0.79	0.53, 1.20	0.269			
**Emesis**	32 (7.7%)	16 (9.9%)	576	1.31	0.68, 2.42	0.409			
**Headache**	50 (12%)	16 (9.9%)	576	0.80	0.43, 1.42	0.450			
**Runny nose**	10 (2.4%)	5 (3.1%)	576	1.29	0.40, 3.68	0.655			
**Nasal congestion**	6 (1.4%)	2 (1.2%)	576	0.85	0.12, 3.73	0.841			
**Sore throat**	112 (27%)	45 (28%)	576	1.04	0.69, 1.55	0.861			
**Myalgia and arthralgia**	38 (9.2%)	15 (9.3%)	576	1.01	0.52, 1.85	0.976			
**Diarrhea**	38 (9.2%)	13 (8.0%)	576	0.86	0.43, 1.63	0.658			
**Chills**	1 (0.2%)	4 (2.5%)	576	10.5	1.53, 205	0.015	22.8	2.18, 554	0.016
**Adynamia**	20 (4.8%)	6 (3.7%)	576	0.76	0.27, 1.82	0.550			
**Odynophagy**	0 (0%)	0 (0%)							
**Anosmia**	33 (9.1%)	5 (3.4%)	512	0.35	0.12, 0.83	0.016	0.35	0.07, 1.53	0.19
**Ageusia**	29 (8.0%)	5 (3.4%)	512	0.40	0.13, 0.97	0.042	1.09	0.22, 5.27	0.92
**Dyspnea**	290 (70%)	136 (84%)	576	2.24	1.42, 3.64	<0.001	2.27	1.34, 3.98	0.003
**Conjunctivitis**	0 (0%)	0 (0%)							
**Diabetes**	67 (16%)	42 (26%)	576	1.81	1.16, 2.80	0.009	1.78	1.04, 3.04	0.034
**Immunodeficiency**	4 (1.0%)	5 (3.1%)	576	3.26	0.85, 13.3	0.082	4.50	0.91, 27.3	0.074
**Heart diseases**	73 (18%)	42 (26%)	576	1.63	1.06, 2.51	0.028	1.63	0.94, 2.81	0.079
**Lung diseases**	1 (0.2%)	3 (1.9%)	576	7.79	0.99, 158	0.051	3.50	0.42, 72.7	0.29
**Neurological diseases**	1 (0.2%)	2 (1.2%)	576	5.16	0.49, 112	0.165	2.12	0.16, 50.9	0.57
**Kidney diseases**	9 (2.2%)	4 (2.5%)	576	1.14	0.31, 3.55	0.832			
**Obesity**	12 (2.9%)	9 (5.6%)	576	1.97	0.79, 4.75	0.141	2.32	0.81, 6.79	0.12
**Asthma**	2 (0.5%)	0 (0%)	576	0.00		0.250	0.00		0.99
**Blood diseases**	2 (0.5%)	0 (0%)	576	0.00		0.250	0.00		0.99
**Liver diseases**	2 (0.5%)	0 (0%)	576	0.00		0.250	0.00		0.99

^1^ Median (IQR); n (%); ^2^ OR = Odds Ratio, CI = Confidence Interval; ^3^ aOR = Adjusted Odds Ratio.

**Table 5 ijerph-20-04639-t005:** Sociodemographic factors, symptoms and comorbidities as predictors of mortality.

Characteristic	Recovered N = 15,779 ^1^	Dead N = 359 ^1^	N	OR ^2^	95% CI ^2^	*p*-Value	aOR ^3^	95% CI ^3^	*p*-Value
**Age**	38 (27, 50)	70 (58, 80)	16,138	1.09	1.08, 1.10	<0.001	1.07	1.06, 1.08	<0.001
**Municipality**			16,138			<0.001			
Altamira	6403 (41%)	188 (52%)		—	—		1.24	0.63, 2.64	0.56
Anapu	1112 (7.0%)	25 (7.0%)		0.77	0.49, 1.14		2.76	1.21, 6.56	0.018
Brasil Novo	790 (5.0%)	22 (6.1%)		0.95	0.59, 1.45		1.15	0.46, 2.93	0.77
Medicilândia	1517 (9.6%)	20 (5.6%)		0.45	0.27, 0.70		1.39	0.57, 3.49	0.48
Pacajá	1581 (10%)	20 (5.6%)		0.43	0.26, 0.67		0.37	0.14, 0.99	0.045
Porto de Moz	1170 (7.4%)	28 (7.8%)		0.82	0.53, 1.20		0.65	0.26, 1.65	0.36
Senador José Porfírio	635 (4.0%)	15 (4.2%)		0.80	0.45, 1.32		0.55	0.17, 1.72	0.31
Uruará	1205 (7.6%)	19 (5.3%)		0.54	0.32, 0.84		1.61	0.65, 4.09	0.30
Vitória do Xingu	1366 (8.7%)	22 (6.1%)		0.55	0.34, 0.84		—	—	
**Sex**			16,138			<0.001			
Male	7358 (47%)	215 (60%)		—	—		—	—	
Female	8421 (53%)	144 (40%)		0.59	0.47, 0.72		1.05	0.76, 1.46	0.77
**Local case**	14,998 (95%)	339 (94%)	16,138	0.88	0.57, 1.44	0.599			
**Imported case**	242 (1.5%)	21 (5.8%)	16,137			<0.001	0.29	0.11, 0.75	0.013
**Fever**	9687 (61%)	254 (71%)	16,138	1.52	1.21, 1.92	<0.001	1.54	1.07, 2.24	0.022
**Cough**	8901 (56%)	255 (71%)	16,138	1.89	1.51, 2.39	<0.001	0.95	0.66, 1.38	0.78
**Emese**	489 (3.1%)	22 (6.1%)	16,137	2.04	1.28, 3.10	0.004	1.34	0.62, 2.77	0.44
**Headache**	5661 (36%)	43 (12%)	16,138	0.24	0.17, 0.33	<0.001	0.81	0.51, 1.28	0.38
**Runny nose**	1550 (9.8%)	13 (3.6%)	16,137	0.34	0.19, 0.58	<0.001	1.44	0.67, 2.85	0.32
**Nasal congestion**	248 (1.6%)	6 (1.7%)	16,137	1.06	0.42, 2.20	0.882			
**Sore throat**	6817 (43%)	98 (27%)	16,137	0.49	0.39, 0.62	<0.001	0.57	0.39, 0.82	0.003
**Myalgia and arthralgia**	3662 (23%)	34 (9.5%)	16,138	0.35	0.24, 0.49	<0.001	0.79	0.47, 1.30	0.36
**Diarrhea**	1385 (8.8%)	26 (7.2%)	16,137	0.81	0.53, 1.19	0.295			
**Chills**	195 (1.2%)	3 (0.8%)	16,137	0.67	0.17, 1.78	0.470			
**Adynamia**	594 (3.8%)	16 (4.5%)	16,137	1.19	0.69, 1.92	0.508			
**Odynophagy**	70 (0.4%)	0 (0%)	16,137	0.00	0.00, 0.11	0.076	0.00		0.98
**Anosmia**	3193 (21%)	11 (3.4%)	15,666	0.13	0.07, 0.23	<0.001	0.31	0.10, 0.89	0.035
**Ageusia**	2972 (19%)	12 (3.7%)	15,666	0.16	0.08, 0.27	<0.001	0.86	0.28, 2.33	0.78
**Dyspnea**	4236 (27%)	287 (80%)	16,137	10.9	8.42, 14.2	<0.001	3.66	2.57, 5.26	<0.001
**Conjunctivitis**	27 (0.2%)	0 (0%)	16,137	0.00	0.00, 15.2	0.270			
**Diabetes**	305 (1.9%)	76 (21%)	16,137	13.6	10.3, 17.9	<0.001	1.01	0.60, 1.68	0.98
**Immunodeficiency**	120 (0.8%)	5 (1.4%)	16,137	1.84	0.65, 4.09	0.223	0.40	0.07, 1.58	0.25
**Heart diseases**	428 (2.7%)	86 (24%)	16,137	11.3	8.67, 14.6	<0.001	2.05	1.26, 3.30	0.004
**Lung diseases**	66 (0.4%)	3 (0.8%)	16,137	2.01	0.49, 5.43	0.288			
**Neurological diseases**	2 (<0.1%)	2 (0.6%)	16,137	44.2	5.29, 369	0.002	0.91	0.01, 110	0.98
**Kidney diseases**	28 (0.2%)	9 (2.5%)	16,137	14.5	6.40, 29.7	<0.001	3.31	0.83, 12.1	0.079
**Obesity**	17 (0.1%)	16 (4.5%)	16,137	43.2	21.5, 86.7	<0.001	2.76	0.86, 9.49	0.10
**Asthma**	30 (0.2%)	2 (0.6%)	16,137	2.94	0.47, 9.78	0.204	3.17	0.33, 19.0	0.26
**Blood diseases**	3 (<0.1%)	2 (0.6%)	16,137	29.5	3.87, 178	0.003	14.3	0.23, 883	0.36
**Liver diseases**	3 (<0.1%)	2 (0.6%)	16,137	29.5	3.87, 178	0.003			
**Clinical management**			16,137			<0.001			
Home care	15,472 (98%)	89 (25%)		—	—		—	—	
Medical ward	274 (1.7%)	140 (39%)		88.8	66.5, 119		21.1	14.5, 30.8	<0.001
ICU	32 (0.2%)	130 (36%)		706	461, 1112		258	146, 474	<0.001
Unknown	1	0							

^1^ Median (IQR); n (%); ^2^ OR = Odds Ratio, CI = Confidence Interval; ^3^ aOR = Adjusted Odds Ratio, CI = Confidence Interval.

## Data Availability

All data were made available by the https://www.covid-19.pa.gov.br/#/ (accessed on 31 August 2022).

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
