# Peer review of "COVID-19 in the Eastern Brazilian Amazon: Incidence, Clinical Management, and Mortality by Social Determinants of Health, Symptomatology, and Comorbidities in the Xingu Health Region"

_ijerph, 2023, doi:10.3390/ijerph20054639_

Round 1

Reviewer 1 Report

The presentation of the region for which the research of the study was carried out it is welcome, as it is part of the Northeast area of ​​Brazil, the country that was, after the United States of America, the hardest hit by the the SARS-CoV-2 pandemic in that period of time.

In my opinion, if the presented study took place from March 2020 to March 2021, in the introduction part, I do not find necessary to mention data about the incidence of infections and mortality from August 31, 2022. (page2/26).

The article is interesting but rather difficult to follow because the information from the tables in the results section are repeated in the text, therefore I recommend that you avoid describing the data from the tables in the text. Or if you leave them in text format, remove them from the tables.

I found 2 mistyping,  in Keywords: "pademic" and on page 21..."inin-angiotensin-..."

Author Response

Response to Reviewer 1 Comments

Point 1: In my opinion, if the presented study took place from March 2020 to March 2021, in the introduction part, I do not find necessary to mention data about the incidence of infections and mortality from August 31, 2022. (page2/26).

Response 1: Thanks for the comment. As required, the information has been withdrawn.

Point 2: The article is interesting but rather difficult to follow because the information from the tables in the results section are repeated in the text, therefore I recommend that you avoid describing the data from the tables in the text. Or if you leave them in text format, remove them from the tables.

Response 2: Thanks for the comment. The data has been presented in tables and removed from the text, as required.

Point 3: I found 2 mistyping,  in Keywords: "pademic" and on page 21..."inin-angiotensin-..."

Response 3: Thanks for your comment. The two words have been corrected.

Reviewer 2 Report

Lima Figueiredo et al. aimed to describe incidence, clinical management and mortality by social determinants of health, symptomatology and comorbidities among pateints with COVID-19 in in eastern Brazilian Amazon. This kind of paper still have a big interest and is also treating a ‘hot topic’. However, there are some issues to address:

1.       Introduction: introduction is quite poor regarding SARS-CoV-2 variety of clinical features and history. It is just a geographical description, which can be reduced favoring other aspects. Please, carefully revise your introduction following the scheme as below:

-          At the end of December 2019, a new respiratory syndrome was described in Wuhan, China (http://wjw.wuhan.gov.cn/front/web/showDetail/2019123108989)

-          After detection, it was defined as SARS-CoV.2 (http://www.xinhuanet.com/english/2020-01/09/c_138690570.htm)

-          After a few months, WHO declared it a major concern for Public health (https://www.who.int/docs/default-source/coronaviruse/situation-reports/20200729-covid-19-sitrep-191.pdf?sfvrsn=2c327e9e_2)

-          COVID-19 has a wide spectrum of clinical features, such as fever, cough, and dyspnea. Instead, less frequent symptoms are the smell and taste alterations, gastrointestinal symptoms, headache, and skin rash. (Read and use these articles: https://doi.org/10.1002/hed.26204,   https://doi.org/10.26355/eurrev_202007_22291, https://doi.org/10.1016/S1473-3099(20)30402-3, https://doi.org/10.1016/S0140-6736(20)30183-5,  https://doi.org/10.1111/jdv.16669;)

-          Since the start of pandemic till now, literature on possible treatments is increasing (doi: 10.7573/dic.2020-10-3.; doi: 10.3390/healthcare10050956.)

-          However, the best care is still unknown in some diseases’ phases (e.g. non invasive ventilation etc.)

-          Please, describe the negative effect of SARS-CoV-2 on the clinical practice and peoples’ health and wellbeing (https://www.who.int/emergencies/diseases/novel-coronavirus-2019?adgroupsurvey={adgroupsurvey}&gclid=CjwKCAiAjs2bBhACEiwALTBWZVX8ZrIcaJ1U0hr6yifNDw7hnhmcSkU__HkyOunI79judSvmYfbsNRoCzVYQAvD_BwE; doi: 10.1159/000518490; DOI: 10.1016/j.jad.2020.08.001.; DOI: 10.1136/bmjopen-2020-042871.)

-          Then, state the geographical area you are studying

2.       Methods: please, reformulate the structure, it is extremely messy. Follow the order as below

2.1 Study design

2.2 Population and study time

2.3 Inclusion and exclusion criteria

2.4 Setting

2.5 Assessments (instead of database)

2.5 Statistical analysis

2.6 Ethical issues

      3.  Results: I gave some advice in the attached text.

      4. English language: the text is full of typos and grammar mistakes (that instead of which and not only). Please, refer to a native speaker before the resubmission

Author Response

Response to Reviewer 2 Comments

Point 1:  Introduction is quite poor regarding SARS-CoV-2 variety of clinical features and history. It is just a geographical description, which can be reduced favoring other aspects. Please, carefully revise your introduction following the scheme as below: 

Response 1: Thanks for the comment. As required, new information has been added in the introduction based on suggested references. Therefore, all your suggestions were accepted.

  • At the end of December 2019, a new respiratory syndrome was described in Wuhan, China (http://wjw.wuhan.gov.cn/front/web/showDetail/2019123108989)
  • Response: Please see the first paragraph of the introduction.

  • After detection, it was defined as SARS-CoV.2 (http://www.xinhuanet.com/english/2020-01/09/c_138690570.htm)
  • Response: Please see the first paragraph of the introduction.

  • After a few months, WHO declared it a major concern for Public health (https://www.who.int/docs/default-source/coronaviruse/situation-reports/20200729-covid-19-sitrep-191.pdf?sfvrsn=2c327e9e_2)
  • Response: Please see the first paragraph of the introduction.

  • COVID-19 has a wide spectrum of clinical features, such as fever, cough, and dyspnea. Instead, less frequent symptoms are the smell and taste alterations, gastrointestinal symptoms, headache, and skin rash. (Read and use these articles: https://doi.org/10.1002/hed.26204,   https://doi.org/10.26355/eurrev_202007_22291, https://doi.org/10.1016/S1473-3099(20)30402-3, https://doi.org/10.1016/S0140-6736(20)30183-5,  https://doi.org/10.1111/jdv.16669;)
  • Response: Please see the second paragraph of the introduction.

  • Since the start of pandemic till now, literature on possible treatments is increasing (doi: 10.7573/dic.2020-10-3.; doi: 10.3390/healthcare10050956.)
  • Response: Please see the third paragraph of the introduction.

  • However, the best care is still unknown in some diseases’ phases (e.g. non invasive ventilation etc.)
  • Response: Please see the fourth paragraph of the introduction.

  • Please, describe the negative effect of SARS-CoV-2 on the clinical practice and peoples’ health and wellbeing (https://www.who.int/emergencies/diseases/novel-coronavirus-2019?adgroupsurvey={adgroupsurvey}&gclid=CjwKCAiAjs2bBhACEiwALTBWZVX8ZrIcaJ1U0hr6yifNDw7hnhmcSkU__HkyOunI79judSvmYfbsNRoCzVYQAvD_BwE; doi: 10.1159/000518490; DOI: 10.1016/j.jad.2020.08.001.; DOI: 10.1136/bmjopen-2020-042871.)
  • Response: Please see the fourth paragraph of the introduction.

  • Then, state the geographical area you are studying
  • Response: The contextualization of the previous introduction was maintained.

Point 2:   Methods: please, reformulate the structure, it is extremely messy. Follow the order as below

2.1 Study design

2.2 Population and study time

2.3 Inclusion and exclusion criteria

2.4 Setting

2.5 Assessments (instead of database)

2.5 Statistical analysis

2.6 Ethical issues

Response 2: Thanks for your comment. As required, the Methods structure was reformulated following the informed order.

Point 3: Results: I gave some advice in the attached text.

Response 3: Thanks for your comment. As required, the changes indicated in the pdf file have been made. However, only the presentation in Table 1 was maintained because we believe it would impair the quality of the reading. Additionally, all other requests in the attached file have been modified.

Point 4: English language: the text is full of typos and grammar mistakes (that instead of which and not only). Please, refer to a native speaker before the resubmission.

Response 4: We thank you for the appointments. The text underwent a new revision for typographical errors, as requested.  We inform you that the manuscript was submitted to a certified company for review in English. 

Round 2

Reviewer 2 Report

The authors substantially answered all my questions.